# Potential diagnostic markers and therapeutic targets for DM2 and periodontitis based on bioinformatics analysis

**Rong Luo** *, **Zhenye Liang, Huijun Chen, Dandan Bao, Xinlu Lin**

Department of Pharmacy, Taizhou First People's Hospital, Taizhou, China

* 13736600301@163.com

## Abstract

### Background

Diabetes Mellitus type 2 (DM2) is thought to have a bidirectional relationship with Periodontitis (PD). However, the complex molecular interactions between DM2 and PD remain unclear. This study aimed to explore the shared genes and common signatures of DM2 and PD via bioinformatic analysis.

### Methods

Firstly, using bioinformatic methods to investigate common genes. The series matrix files of *GSE6751* for DM and *GSE15932* for PD were downloaded from the Gene Expression Omnibus (GEO) database. The data was normalized using the R package, and the limma package was utilized to identify the Differentially Expressed Genes (DEGs). Gene Ontology and Kyoto Encyclopedia of Genes and Genomes enrichment analyses of DEGs were performed using the "clusterProfiler" package in the R software. The protein-protein network was constructed to analyze the potential relationship among the proteins. CytoHubba, a plugin for the Cytoscape software, was used to identify the hub genes. The validation datasets selected for DM2 and PD were *GSE10334* and *GSE7014*, respectively. Receiver Operating Characteristic (ROC) curve analysis was performed to obtain the area under the ROC curve. Lipopolysaccharide (LPS) + high glucose-induced DM-related PD was simulated to verify the three hub genes through quantitative Real-Time Polymerase Chain Reaction (qRT-PCR) and Western blot (WB).

### Results

In total, 44 common DEGs were identified. ITGAM, H2BC21, S100A9 was identified as he hub genes of DM2 and PD, with all of them were up-regulated. In addition, the area under the curve of all three hub genes was more than 0.65. In-vitro experiments revealed that the relative expression of *S100A9* was increased after the treatment with LPS + high glucose. Besides, TLR4 and p-NF-κB levels were also improved in model group.

**Data availability statement:** All data are available in the manuscript and/or supporting information files.

**Funding:** This research was supported by Taizhou Science and Technology planning Project (NO. 20ywb67). Huijun Chen is the funder who analyzed the data in this manuscript.

**Competing interests:** The authors have declared that no competing interests exist.

## Conclusion

*S100A9* was identified as the hub gene of DM2 and PD. S100A9 could trigger TLR4 signaling way to promote disease development, which can be the potential targets for diagnosis and treatment.

## Introduction

Periodontitis (PD) is a chronic multifactor inflammatory disease with a prevalence of 45–50%. It is characterized by progressive destruction of the tooth-supporting apparatus and can be treated with dysbiotic plaque biofilm control [1]. Various studies support that periodontitis is independently associated with several non-communicable diseases, including Diabetes Mellitus (DM) [2]. The sixth complication of diabetes, PD, has been identified, with individuals diagnosed with type 2 diabetes (DM2) being at a nearly two-fold higher risk of developing this condition compared to those without diabetes [3]. However, the underling mechanism between the two disease was unclear.

Numerous studies leveraged the constant advancements in bioinformatics to excavate underlying mechanisms for several diseases [4]. It is imperative to further investigate the potential biological mechanisms underlying intricate relationship between PD and DM2. Therefore, this study was composed with two parts, firstly downloading periodontitis and DM2 datasets from the Gene Expression Omnibus (GEO) database and identified the Differentially Expressed Genes (DEGs) using the limma package. Secondly, the common hub genes of the two diseases were investigated to elucidate the interrelated mechanisms between periodontitis and DM2. Finally, in-vitro experiments were performed to verify the hub genes of DM2 and periodontitis. The detailed schematic workflow of this part is shown in Fig 1.

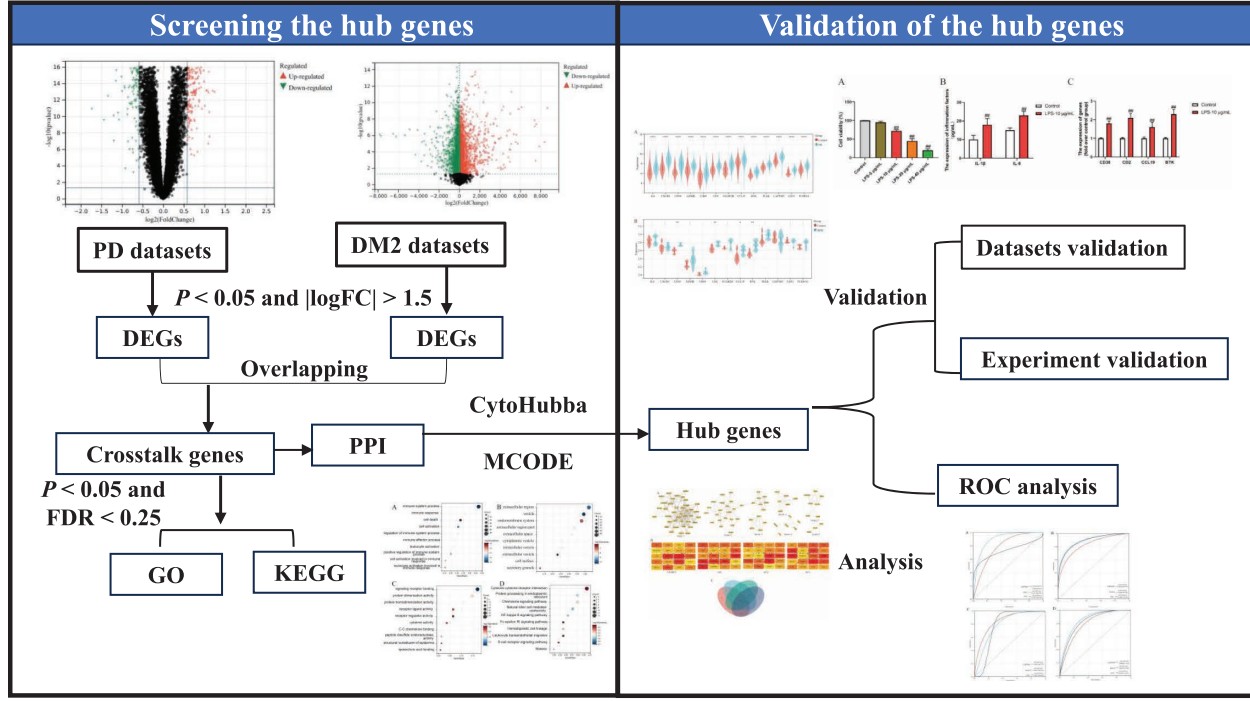

**Fig 1. Workflow.**

## Methods and materials

### Data downloading and preprocessing

We utilized GEO (https://www.ncbi.nlm.nih.gov/), a public functional genomics database, to obtain periodontitis and DM2 datasets by considering "periodontitis" or "type 2 diabetes" as the keywords. The inclusion criteria were as follows: a. the study included both diabetic and healthy control samples, b. the samples were of *Homo sapiens*, and c. the chip data type was a gene expression profile. According to the criteria, two datasets, GSE6751 [5] and GSE15932 [6], were included in this study. GSE6751 consisted of Peripheral Blood Mononuclear Cells (PBMC) samples from 15 PD and 15 controls; GSE15932 included PBMC samples from 8 DM2 and 8 controls.

### Identification of DEGs

The series matrix files for *GSE6751* and *GSE15932* were downloaded from GEO. The R package was utilized to normalize the data, and the limma package was used to identify the DEGs of PD/DM2 and control samples. Specifically, an expression profile dataset was obtained, and log2 transformation was applied to normalize the data. Subsequently, the lmFit function was employed for multiple linear regression analysis, followed by the eBayes function to compute moderated t-statistics, moderated F-statistics, and log-odds of differential expression through empirical Bayes moderation of standard errors towards a common value. Ultimately, the significance of differential expression for each gene was determined. Genes meeting the specific cutoff criteria of $P < 0.05$ and $|logFC| > 1.5$ were defined as DEGs. An online Venn diagram tool was used to map the common DEGs (http://www.sangerbox.com/home.html).

### Functional and pathway enrichment analysis

In order to analyze the function of the common DEGs, Gene Ontology (GO) annotation and Kyoto Encyclopedia of Genes, including Biological Process (BP), Cellular Component (CC) Molecular Function (MF), and Kyoto Encyclopedia of Genes and Genomes (KEGG) pathway enrichment analysis of DEGs were performed via the "cluster profiler" package in R software [7]. "Homo Sapiens" was set as the species and the top 10 items were selected to draw the bubble map.

### PPI network construction and identification of hub genes

The common DEGs were imported into the Search Tool for the Retrieval of Interacting Genes (STRING) database (https://cn.string-db.org/) to construct the Protein-Protein Interaction (PPI) network. Confidence scores were set to intermediate values (>0.4). CytoHubba, a plugin for the Cytoscape software, was used to explore the important nodes of the biological network. In addition, Molecular Complex Detection (MCODE), a plugin for Cytoscape, was used to filter the significant modules of core genes from the PPI network complex. The criteria were as follows: Degree Cutoff = 2, Node Score Cutoff = 0.2, K-Core = 2, and Max. Depth = 100.

### Validation of hub gene expression

The expression levels of the identified hub genes were validated in *GSE10334* and *GSE7014* for PD and DM2, respectively. *GSE10334* included 183 samples periodontitis and 64 healthy control gingival tissue samples. *GSE7014* contained the Ribonucleic acid (RNA)-sequencing results of skeletal muscle biopsies from 20 DM2, and six healthy control samples. The rank sum test was used to compare the two databases, with $P < 0.05$ considered statistically significant.

## Receiver operating characteristic curves

The Receiver Operating Characteristic (ROC) curve analysis was conducted using the R software package pROC (version 1.17.0.1) to obtain the area under the ROC curve. Further, the gene expression scores for *ITGAM, H2BC21 and S100A9* were calculated, and the final Area Under the Curve (AUC) result was obtained by evaluating the AUC and confidence interval using the ci function of pROC.

## Regents

Dulbecco's Modified Eagle's Medium (DMEM, 11995) and 10% Fetal Bovine Serum (FBS, A8020) were purchased from Solarbio (Beijing, China). We sourced Penicillin-Streptomycin Solution (C0222), Sodium Pyruvate (113-24-6), L-glutamine (C0212), and Thiazolyl Blue Tetrazolium Bromide Assay Kit (ST1537) from Beyotime (Shanghai, China). Human periodontal ligament cells (PDLCs, CS-2071X) were obtained from Shanghai C-reagent Biotechnology Co. Ltd (Shanghai, China). Lipopolysaccharide (LPS, L5293-2ML) was acquired from Sigma-Aldrich. TRIzol solution was obtained from Invitrogen (CA, USA). MonScript™ RTIII All-in-One Mix with dsDNase (Lot.430530) was purchased from Monad (Wuhan, China). The Enzyme-linked Immunosorbent Assay (ELISA) kits of IL-1β (ml058059) and IL-6 (ml028583) were procured from Shanghai Enzyme-linked Biotechnology Co., Ltd. (Shanghai, China). The antibodies of S100A9 (26992-1-AP), TLR4 (19811-1-AP), NF-κB (10745-1-AP), P-NF-κB (82335-1-RR) were obtained from Proteintech (Wuhan, China).

## Cell treatment

We seeded PDLCs in DMEM (containing 10% FBS, 100 U/mL penicillin, 100 μg/mL streptomycin, 1 mmol/L sodium pyruvate, and 2 mmol/L L-glutamine) and cultured them in an incubator at 37°C with 5% $CO_2$. The PDLCs were divided into two groups: the control group (with 8 mmol/L glucose) and the model group (with 25 mmol/L glucose and 5, 10, 20, and 40 μg/mL concentrations of LPS).

## MTT assay

The PDLCs were seeded in a 96-well plate at the density of $1 \times 10^5$ cells/mL and cultured in an incubator at 37°C and 5% $CO_2$ for 24 hours. Following this treatment, we incubated the PDLCs in the model group with LPS for another 24 hours, and the control group was replaced by DMEM. Further, we added 50 μL MTT (2.5 mg/mL) into the two group cells and continued incubation for another four hours. Finally, after adding DMSO (Dimethyl Sulfoxide) (200 μL), the OD (Optical Density) value was read at a wavelength of 490 nm.

## Enzyme-linked immunosorbent assay

After completing the above treatment, we obtained the supernatant from the PDLCs and determined the levels of IL-1β and IL-6 as per the prescribed protocols.

## Quantitative real-time polymerase chain reaction (qRT-PCR)

We extracted the RNA via TRIzol methods and converted it to cDNA (complementary Deoxyribonucleic acid) using MonScript™ RTIII All-in-One Mix with dsDNase. Following this, we performed qRT-PCR using MonAmp™ SYBR® Green qPCR Mix under the following conditions: pre-degeneration at 95°C for 30 s, degeneration at 95°C for 10 s, annealing and extending at 60°C for 30 s. The primer sequences are shown in Table 1. The fold change was calculated using the $2^{-\Delta\Delta Ct}$ method. *GAPDH* was the endogenous reference gene for *ITGAM, S100A9, H2BC21, TLR4, NF-κB*.

**Table 1. Primer sequences.**

| Genes | Forward (5'-3') | Reserve (5'-3') |
|---|---|---|
| *ITGAM* | ACCTGCTCTCCATCCTGCTT | GACGAGTTCACAGGCACTGA |
| *S100A9* | CCCTGAGTGGTGAGTTCAC | ACCTGAGTAGAGGGTGAGG |
| *H2BC21* | ATGGCTGCTGCTGCTGTTG | CGCACGGCACCTTCTCCT |
| *TLR4* | GCAGTCTCCTGGTGCTTTT | GCTCTGGCCACGAAGTCT |
| *NF-κB* | CATGCCGCTGCTCTTCTG | AGGAGGCAGAAAGAAAGGACC |
| *GAPDH* | TGGAGAAACCTGCCAAGTATGA | GGTCCTCAGTGTAGCCCAAG |

## Western blot

Extract proteins from cells using lysis buffer containing protease inhibitors. Quantify the protein concentration using BCA assay. Prepare a polyacrylamide gel, load equal amounts of protein samples and a molecular weight marker into the wells. Run the gel at a constant voltage until the dye front reaches the desired distance. After electrophoresis, transfer proteins from the gel to a PVDF membrane using electroblotting. Incubate the membrane in 5% non-fat milk for 2 hours at room temperature to prevent non-specific binding. Add primary antibodies (S100A9, TLR4, NF-κB, p-NF-*κB)* to target protein and incubate overnight at 4°C. Wash the membrane multiple times with TBST. Apply a labeled secondary antibody and incubate for 1-2 hours at room temperature. Use a chemiluminescent substrate to visualize the bound antibodies. Capture images using a gel documentation system. Compare band intensity against a control to quantify the protein levels by Image J software.

## Statistical analysis

The data is represented as the mean ± SD (Standard Deviation). We conducted the Student's t-test to compare the two groups. $P < 0.05$ was considered statistically significant.

# Results

## Identification of common DEGs

Four expression profiles (GSE6751, GSE15932) were obtained from the GEO database. GSE6751 consisted of Peripheral Blood Mononuclear Cells (PBMC) samples from 15 PD and 15 controls; GSE15932 included PBMC samples from 8 DM2 and 8 controls. The GSE6751 dataset contained 231 DEGs between PD and the control group samples, whereas GSE15932 included 4528 DEGs between DM2 samples from patients with periodontitis and the control group (Fig 2A and B). Venn diagrams revealed the two datasets have 40 up-regulated and 4 down-regulated common genes (Fig 2C and D).

## GO and KEGG Enrichment Analysis

We analyzed the biological information of the common DEGs through GO and KEGG analyses. The results concerning BP depicted that the genes were mainly involved with exocytosis, cell activation, etc. Regarding CC, the genes were enriched in Secretory vesicle, secretory granule, etc. As for MF, the genes were mainly enriched in signaling receptor binding, protein containing complex binding etc. (Fig 3A–C). Besides, KEGG enrichment analysis indicated that Protein processing in PD-L1 expression and PD-1 check point pathway in cancer, Osteoclast differentiation, and so on (Fig 3D).

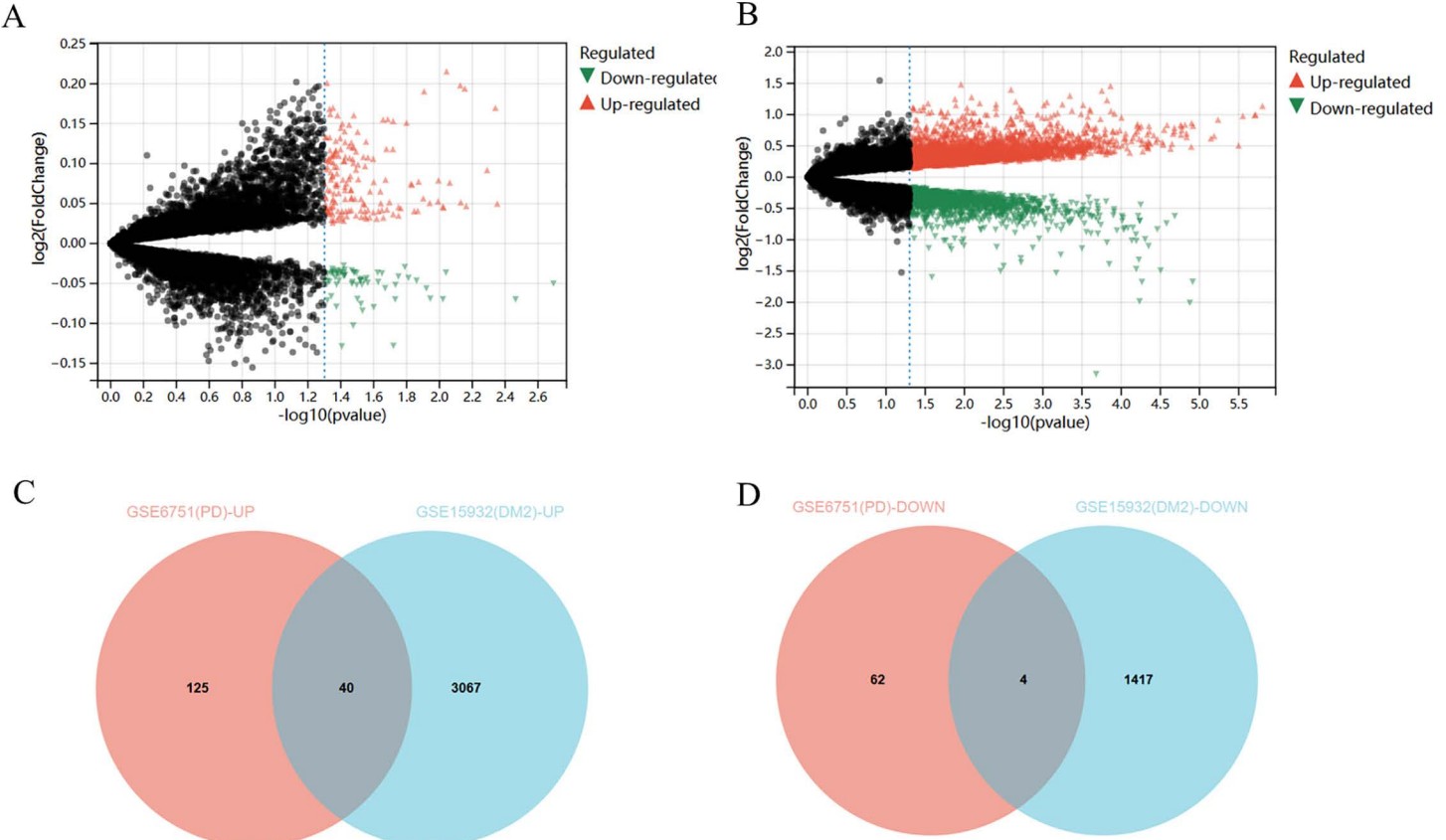

**Fig 2. The common DEGs of GSE6751 and GSE15932.** (A). Volcano map of GSE156993. (B). Volcano map of GSE15932. (C-D). Veen diagrams indicate the common up-regulated and down-regulated genes of the two datasets.

## PPI network analysis of the common DEGs and identification of hub genes

To explore the potential relationship among the proteins coded by these common DEGs, we performed a PPI network analysis, which consisted of 35 nodes and 25 edges with PPI enrichment P-value < 8.82e-05 ( S1 Fig). Only one model was built from the PPI network, which contained 6 nodes, 12 edges, and the cluster score was 4.8 (Fig 4A). Then, Degree, Edge Percolation Component (EPC), Maximum Neighborhood Component (MNC), and Maximal Clique Centrality (MCC) were used to predict the top 10 important hub genes (Fig 4B). As shown in Fig 4C, there were 8 candidate hub genes: *ITGAM, TLR2, FGR, LYN, FOS, S100A9, CXCL16, H2BC21*.

## Validation of hub genes

The validation datasets selected for PD and DM2 were *GSE10334* and *GSE7014*, respectively. The 8 candidate hub genes were all obviously changed between PD and control group (Fig 5A). As shown in Fig 5B, only three genes were significantly different in the DM2, as compared with the control group. Precisely, as expected, *ITGAM, S100A9, H2BC21* were all regulated in PD (*GSE10334*: PD *vs* Control: *ITGAM*: $5.97 \pm 0.03$ *vs* $5.63 \pm 0.30$; *S100A9*: $13.95 \pm 0.56$ *vs* $13.92 \pm 0.22$; *H2BC21*: $7.05 \pm 0.50$ *vs* $7.34 \pm 0.53$, $P < 0.05$) and DM2 groups (*GSE7014*: DM2 *vs* Control: *ITGAM*: $175.01 \pm 15.14$ *vs* $155.71 \pm 19.91$; *S100A9*: $634.75 \pm 287.30$ *vs* $435.37 \pm 46.42$; *H2BC21*: $1400.49 \pm 257.79$ *vs* $1048.82 \pm 378.26$, $P < 0.05$).

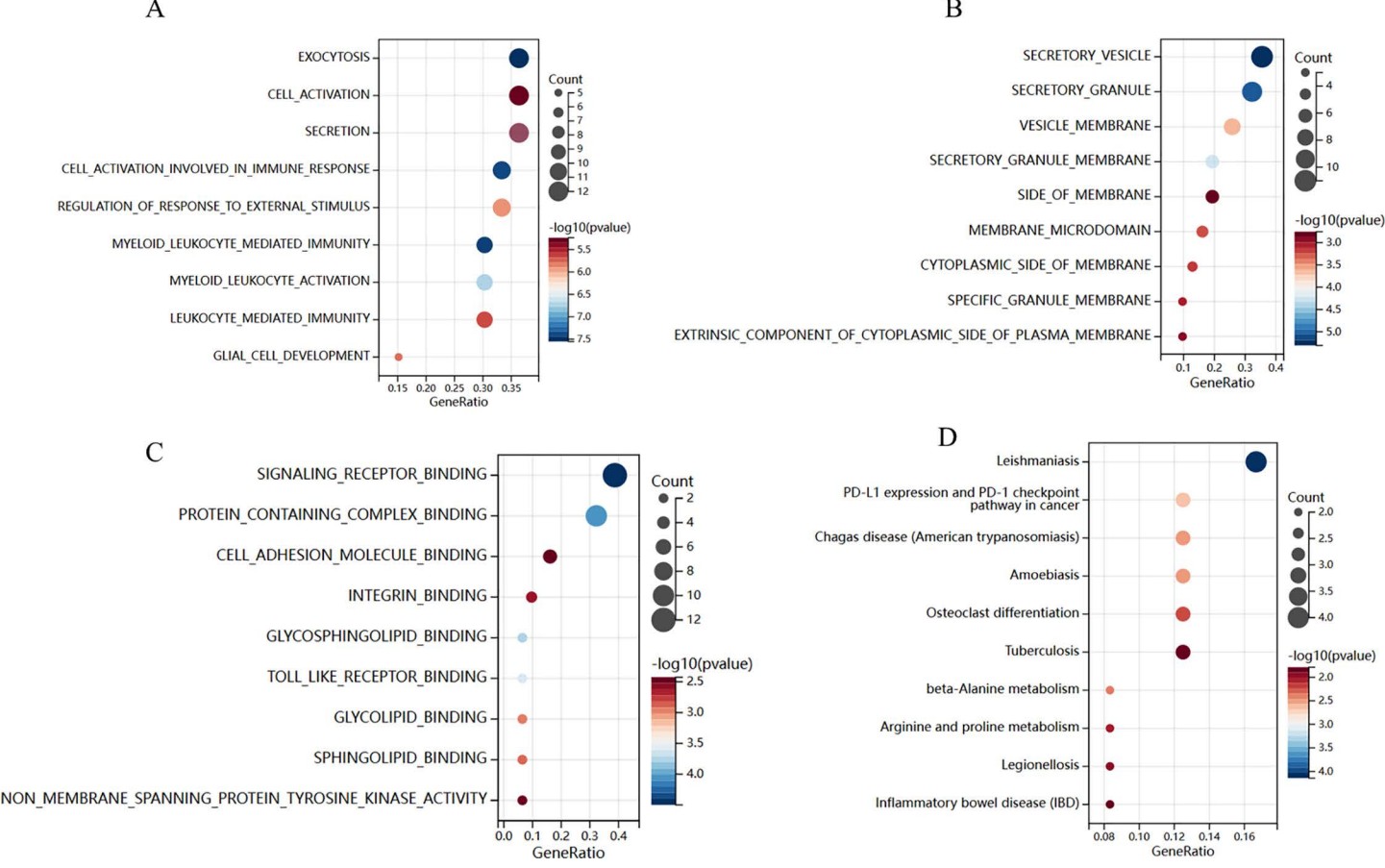

**Fig 3. Functional enrichment analysis of the common DEGs.** GO enrichment analysis included (A) BP, (B) CC and (C) MF, which were selected 10 terms to make bubble diagram. (D) KEGG enrichment analysis was presented by loop graph.

## Efficacy evaluation and PPI construction of hub genes

We calculated the AUC by performing ROC analysis using the R software package pROC (version 1.17.0.1). As shown in Fig 6A–D, the AUC values of all three hub genes were more than 0.65 in *GSE6751, GSE15932, GSE10334*, and *GSE7014*. The results suggest that *ITGAM, H2BC21* and *S100A9* are the potential diagnostic markers of PD and DM2.

The four hub genes and the six genes they interacted with were analyzed using GeneMA-NIA to predict the interactions between the protein network and pathway. The results showed these genes mainly associations with anti-microbial humoral response, pattern recognition receptor signaling pathway, positive regulation of apoptotic process, humoral immune response (Fig 7).

## Verification of the identified hub genes in vitro experiment

We simulated an in-vitro model of DM-related periodontitis induced by LPS in a high-glucose environment. Firstly, we treated the PDLCs with LPS in different doses (5, 10, 20, and 40 μg/mL) and glucose (25 mmol/L) and observed that the LPS dose of 10 μg/mL was optimal for experimentation (Fig 8A). In addition, we also detected the expressions of inflammation cytokines (IL-1β and IL-6) to confirm the occurrence of an inflammatory response. Following the treatment of LPS (10 μg/mL) and maintaining the high glucose state, it was observed that

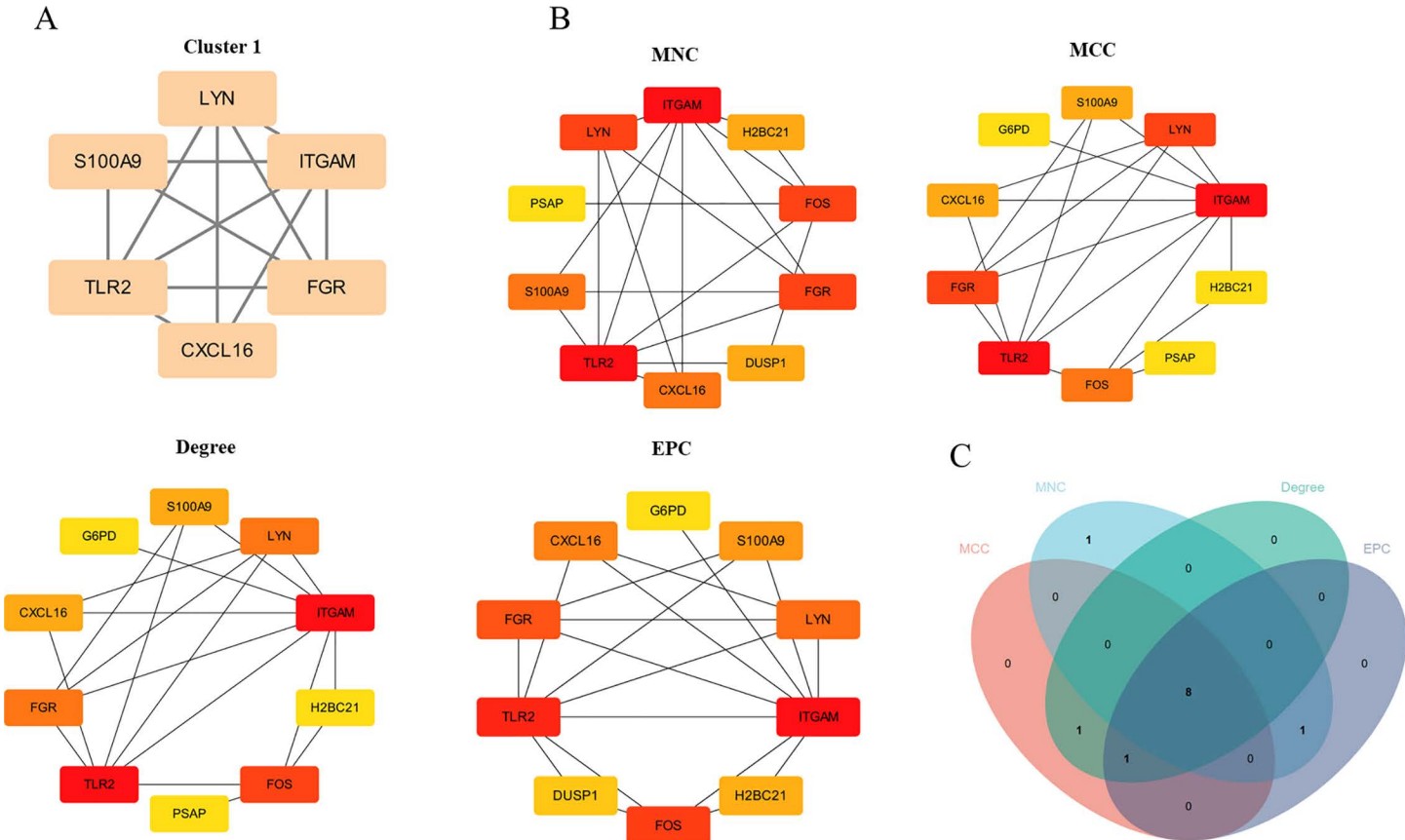

**Fig 4. PPI network analysis of the common DEGs.** (A) Significant gene module and enrichment analysis of the modular genes. (B) Degre, EPC, MNC and MCC were used to predict hub genes. (C) Veen diagrams of the four algorithms.

IL-1β and IL-6 levels increased as expected (*P* < 0.01, Fig 8B). Further, we detected *ITGAM, H2BC21* and *S100A9* levels of DM and PD by qRT-PCR. The relative expressions of only *S100A9* were obviously improved with the induction of LPS and high glucose (*P* < 0.01, Fig 8C). In addition, the WB data also showed the similarly result. The results suggest that *S100A9* could be effective targets for DM2 and PD.

The expression of TLR4 signaling pathway-related proteins was further determined, as TLR4 serves as one of the common receptors for S100A9. The protein and gene expression levels of TLR4 and p-NF-κB in PDLCs cells were significantly upregulated following treatment with LPS+high glucose (*P* < 0.01), as depicted in Fig 9A-C. The above sentence can be polished as follows: "This finding also suggests that A100A9 may facilitate the onset and progression of the disease through activation of the TLR4 signaling pathway

## Discussion

DM2 and periodontitis are both the common, chronic disease [8]. The epidemiological data confirmed that DM2 was the main risk factor of periodontitis and the susceptibility to periodontitis increases about threefold in people with DM2 [9]. And periodontal inflammation negatively affects glycemic control [10]. The complex molecular interactions between DM2 and periodontitis remain unclear. Several studies explored the potential targets and pathways of diseases using bioinformatics analysis [11,12]. Our study is a pioneering effort to analyze

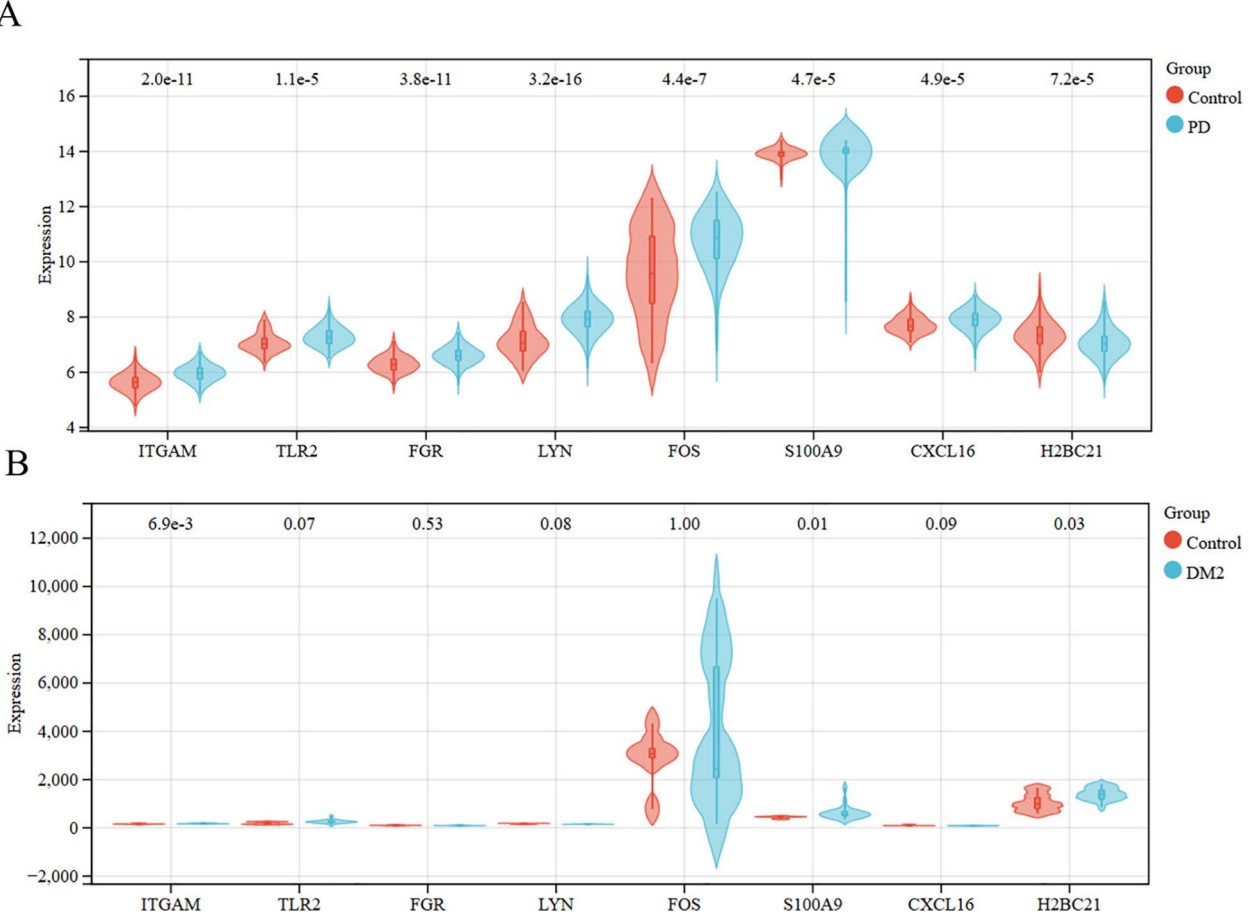

**Fig 5. Validation of the hub genes.** (A) The expressions of hub genes in control and PD groups of GSE10334. (B) The expressions of hub genes in control and DM2 groups of GSE7014. Control group *vs* PD/DM2 group, $P < 0.05$ was considered the significant.

the co-expression genes interrelated with DM2 and periodontitis and establish the association among the hub genes via integrated bioinformatic analyses. It aimed to explore the shared genes and common signatures of DM2 and periodontitis via bioinformatic analysis.

According to the bioinformatic analysis and experiment results, *S100A9* was finally determined as the hub gene, which was up-regulated in DM2 and periodontitis. *S100A9*, the $Ca^{2+}$ binding protein of the S100 family, is preferentially expressed in neutrophils, monocytes, and macrophages, and plays an important role in various inflammatory diseases [13]. Clinically data showed that S100A9 in newly-diagnosed DM2 patients was highly expressed compared with control group [14]. Interestingly, it has been also found that the mRNA levels of S100A9 are significantly upregulated in clinical patients with PD [15,16]. This finding is consistent with our results obtained from bioinformatics analysis, and we also observed a significant upregulation of S100A9 in the model of high-sugar-induced periodontitis that was established.

GeneMANIA result revealed that anti-microbial humoral response, humoral immune response could be the common potential mechanism of PD and DM2. S100A9 is a well-documented antimicrobial and participated in various diseases [17]. Periodontitis is a bacterial infectious disease, and various inflammatory cytokines regulate the pathophysiology of this condition [18]. A study indicated [19] that by ligating the gingival tissue adjacent to the teeth of mice, a murine model of periodontitis was successfully established. Subsequent

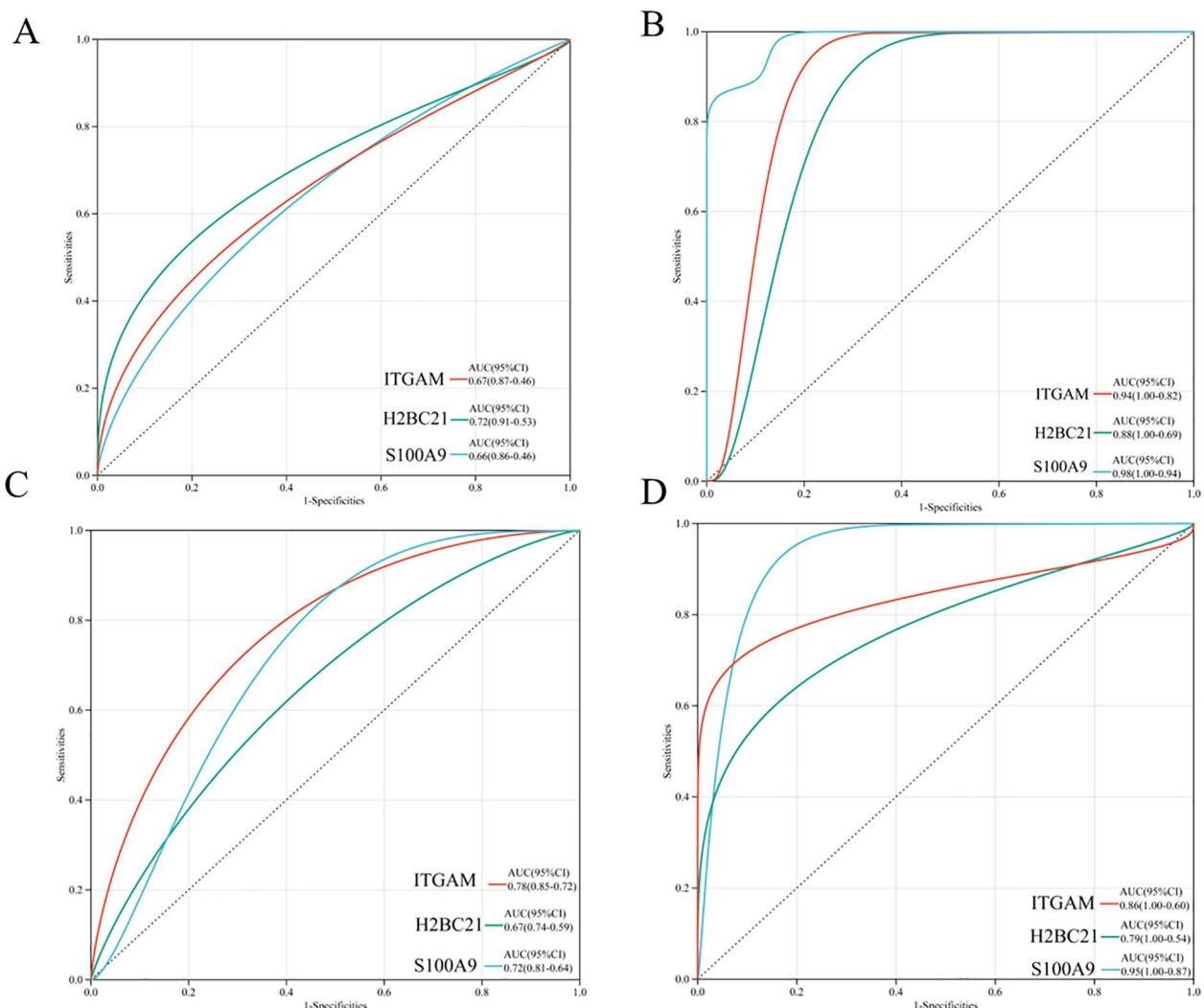

**Fig 6. ROC analysis of the three hub genes.** ROC analysis of ITGAM, H2BC21 and S100A9 of (A) GSE6715 (B) GSE15932, (C) GSE10334 and (D) GSE7014.

analysis revealed significant disparities in oral microbiota composition between wild-type mice and S100A9$^{-/-}$ mice, with notable reduction in alveolar bone resorption observed specifically in S100A9$^{-/-}$ mice. Notably, patients with DM2, characterized by pancreatic β-cell damage, could polarize macrophages towards M1 macrophages to induce an inflammatory to improve insulin resistance, and strengthen ROS activity to increase oxidative response, which may jointly exacerbate T2DM development [20]. The findings suggest that S100A9 has the potential to serve as a promising biomarker for the diagnosis of PD and DM2.

The release of S100A9 by neutrophils is an important factor in triggering inflammation and immune processes. Toll-like receptor 4 (TLR4) is described as the receptor of S100A9 [21], so we further explored the transformation of TLR4 signaling way in the model of high-sugar-induced periodontitis. The data unequivocally demonstrates that in a model of periodontitis induced by high sugar consumption, the activation of TLR4 and subsequent engagement of the downstream factor NF-κB signaling cascade are prominently observed. Nishikawa et al also supported that S100A9 induced IL-6 production via TLR4 signaling way in gingival crevicular

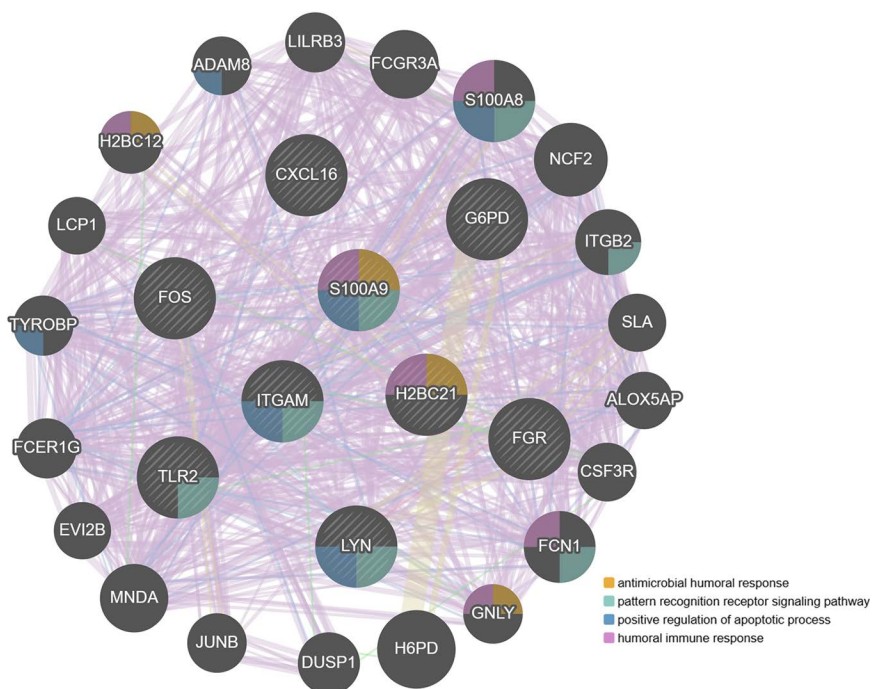

**Fig 7. The gene network of the 4 hub genes along with their interactional genes.**

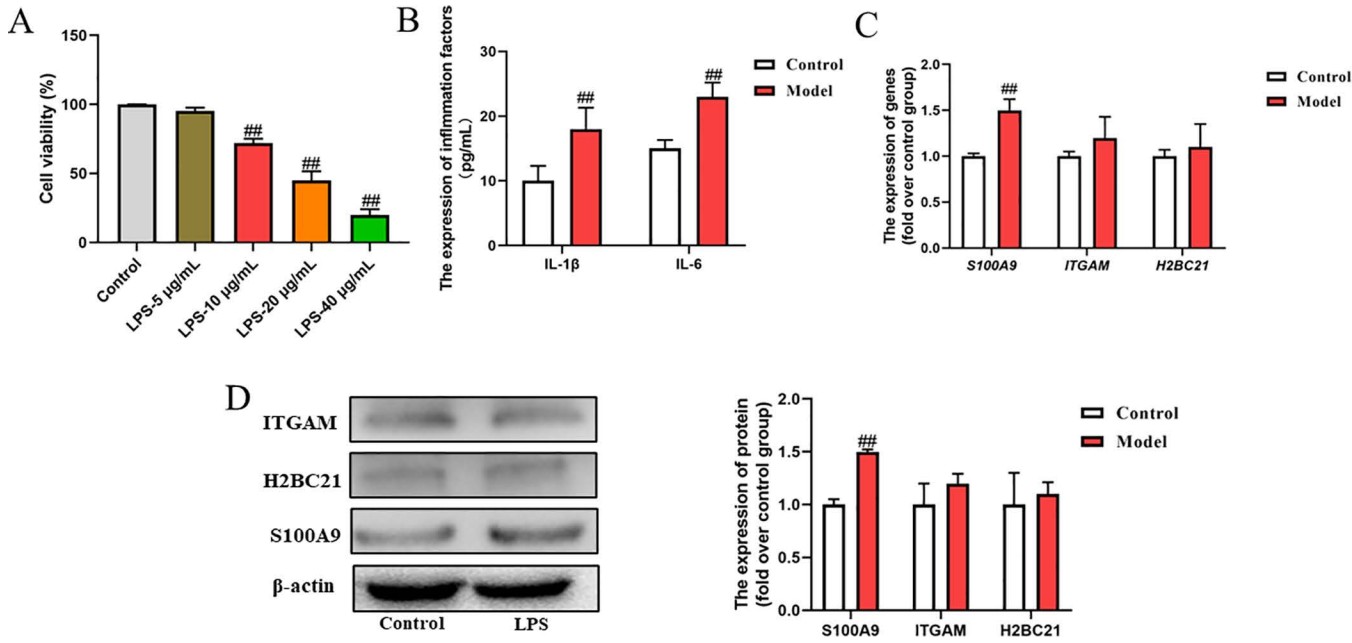

**Fig 8. LPS + high glucose induced an in vitro model of DM related-periodontitis.** (A) MTT assay to detect the cell viability. (B) The expressions of IL-1β and IL-6 after the incubation of LPS and high glucose. (C) The relative expressions of *ITGAM, H2BC21* and *S100A9* by RT-PCR. (D) Western blot is used to measure the ITGAM, H2BC21 and S100A9 expressions. Compared with control group, *P##* < 0.01.

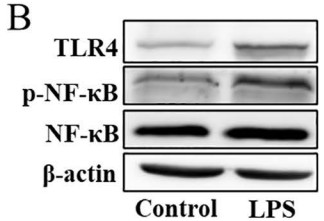
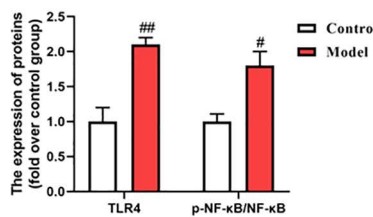

**Fig 9. TLR4 signaling way activation in LPS + high glucose induced model of DM related-periodontitis.** (A) The relative expressions of *TLR4* and *NF-κB* by RT-PCR. (B-C) Western blot is used to measure the TLR4, p-NF-κB and NF-κB expressions. Compared with control group, $P$## < 0.01.

fluids of periodontitis patients [22]. Furthermore, Professor Coppari's research team made the remarkable discovery that the S100A9-TLR4 signaling axis exerts an inhibitory effect on insulin ketogenesis in individuals with diabetes, thereby demonstrating its potential as a highly promising therapeutic target for the treatment of hyperglycemia. This finding holds significant implications for advancing our understanding and management of diabetes [23]. Considering the crucial involvement of S100A9-TLR4 in PD and DM2, it is evident that S100A9 holds significant potential as a valuable diagnostic and therapeutic target for both diseases.

This study presents certain limitations. These are as follows:

a. Even though the role of the hub gene, *S100A9*, in DM2 and PD have been reflected in some clinical and animal research, it has not been validated by laboratory experiments;

b. Our study only utilized the in-vitro cell model and, thus, requires further validation in patients with DM2 and PD, which will be conducted when the required datasets are available.

## Conclusion

This study identified *S100A9* as the hub gene of DM2 and PD. Additionally, S100A9 could trigger TLR4 signaling way to promote disease development, which can serve as potential targets for diagnosis and treatment.

## Supporting information

**S1 Fig. PPI network of DEGs.**
(DOCX)

## Acknowledgment

We acknowledge the GEO database and its contributors for providing valuable datasets. The findings have been presented as a preprint in "Identification of hub genes and biological mechanisms associated with periodontitis and diabetes" following link: https://www. researchsquare.com/article/. Besides, we thank Bullet Edits Limited for the linguistic editing and proofreading of the manuscript. The authors declare that we have not use AI-generated work in this manuscript.

## Author contributions

**Data curation:** Rong Luo.

**Methodology:** Zhenye Liang, Dandan Bao.

**Software:** Huijun Chen, Xinlu Lin.

**Writing – original draft:** Rong Luo.

**Writing – review & editing:** Rong Luo.

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
