## [Decision Letter · Decision Letter 0]

25 Jan 2024

PONE-D-23-39620Identification the hub genes of periodontitis and diabetes by bioinformatics and experimentPLOS ONE

Dear Dr. luo,

Thank you for submitting your manuscript to PLOS ONE. After careful consideration, we feel that it has merit but does not fully meet PLOS ONE’s publication criteria as it currently stands. Therefore, we invite you to submit a revised version of the manuscript that addresses the points raised during the review process.

We look forward to receiving your revised manuscript.

Kind regards,

Qi Zhao

Academic Editor

PLOS ONE

Journal Requirements:

"This research was supported by Taizhou Science and Technology planning Project (NO. 20ywb67)."

5. Please remove your figures from within your manuscript file, leaving only the individual TIFF/EPS image files, uploaded separately. These will be automatically included in the reviewers’ PDF.

Reviewers' comments:

Reviewer's Responses to Questions

**Comments to the Author**

1. Is the manuscript technically sound, and do the data support the conclusions?

Reviewer #1: Yes

Reviewer #2: Yes

2. Has the statistical analysis been performed appropriately and rigorously? 

Reviewer #1: Yes

Reviewer #2: Yes

3. Have the authors made all data underlying the findings in their manuscript fully available?

Reviewer #1: Yes

Reviewer #2: Yes

4. Is the manuscript presented in an intelligible fashion and written in standard English?

Reviewer #1: No

Reviewer #2: Yes

5. Review Comments to the Author

Reviewer #1: 1.English expressions need to be edited more careful and more native, in this manuscript, there are some mistakes. For example, there is no period in the last sentence in method of the abstract.

2. I suggest the authors should add a flowchart in the manuscript to show the process very well.

3. I suggest the authors should elaborate their motivation in the discussion section, as they use several common bioinformatics analysis methods and online tools. What is the novelty and technicality of their work?

4.The advancement of interaction prediction research in various fields of computational biology would provide valuable insights into genetic markers and ncRNAs related with DM, such as miRNA-lncRNA interaction prediction. The authors should discuss it as the future direction. Important computational models in these fields should be cited. Some recommended studies are helpful (PMIDs: 36642414, 36924730, 36305458, 34232474, 37525507, 37660567 and 37466194).

5. The authors should carefully check and unify the information of references. Some references lack the information of volume or contain the wrong page number.

6. Literature review is incomplete in the introduction, especially about the research of differentially expressed genes (DEGs) involvement in periodontitis by computational tools or bioinformatic analysis. I suggest the authors to discuss the recent updates in the related field. Different feature weight calculation methods within a single algorithm result in divergent rankings of DEGs. I think we need consistency and comparable results.

7. Materials and methods section is relatively simple because of no detail about the analysis. The analysis methods and statistical parameters must be clearly emphasized (i.e, reasons for selecting the algorithms used, threshold values used in statistical analysis, etc.).

Reviewer #2: 1.What do “LPS” and “LAPTM5, RAC2, LYN”represent in the Abstract section? What are their full names? When first mentioned, the author should provide their full names to help readers better understand.

2.The authors should add a flowchart in the manuscript to show the process very well.

3.The description of the result is quite simple and too short, especially the descriptions of “Identification of Common DEGs” and “The analysis of immune infiltration”. The authors should add some necessary sentences to describe these results.

4.The labeling of Figures in the paper is quite small. The contents within the figures are unclear. The author needs to carefully revise and modify the figure.

5.The discussion of the deficiencies in current research is quite poor. The authors should discuss it as the future direction. ODE-based theoretical modeling studies on gene/protein signaling networks have been equally important for the study of understanding regulatory mechanisms and finding potential therapeutic targets in diseases (PMID: 35958114, https://doi.org/10.1016/j.chaos.2023.114328, and https://doi.org/10.1103/PhysRevE.108.064412). Would it be possible to discuss and cite these studies in conjunction with the conclusions of this paper?

6.Besides, the advancement of interaction prediction research in various fields of computational biology would provide valuable insights into genetic markers and related diseases. Important computational models in these fields should be discussed and cited. Some recommended studies are helpful (PMIDs: 36584603, 35817399, 36305458).

7.The authors should carefully check and unify the information of references. Some references lack the information of page number, such as refs [7] and [30].

6. PLOS authors have the option to publish the peer review history of their article (what does this mean? ). If published, this will include your full peer review and any attached files.

**Do you want your identity to be public for this peer review?** For information about this choice, including consent withdrawal, please see our Privacy Policy .

Reviewer #1: No

Reviewer #2: No

---

## [Author Response · Author response to Decision Letter 1]

22 Feb 2024

Reviewer #1: 1. English expressions need to be edited more careful and more native, in this manuscript, there are some mistakes. For example, there is no period in the last sentence in method of the abstract.

Reply: Thank you for your suggestion. I have polished my article.

2. I suggest the authors should add a flowchart in the manuscript to show the process very well.

Reply: Thank you for your suggestion. I have added a flowchart in the manuscript.

3. I suggest the authors should elaborate their motivation in the discussion section, as they use several common bioinformatics analysis methods and online tools. What is the novelty and technicality of their work?

Reply: Thank you for your suggestion. I have added to the discussion section about our use of this bioinformatics analysis method and online tools.

4.The advancement of interaction prediction research in various fields of computational biology would provide valuable insights into genetic markers and ncRNAs related with DM, such as miRNA-lncRNA interaction prediction. The authors should discuss it as the future direction. Important computational models in these fields should be cited. Some recommended studies are helpful (PMIDs: 36642414, 36924730, 36305458, 34232474, 37525507, 37660567 and 37466194).

Reply: Thank you very much for your constructive comments. I have carefully studied the relevant literature research provided by you, which will benefit me a lot in my future research methods and methods. And in my discussion part, I introduce the model of relevant literature for discussion.

5. The authors should carefully check and unify the information of references. Some references lack the information of volume or contain the wrong page number.

Reply: Thank you for your suggestion. I have carefully checked and unified the information of references.

6. Literature review is incomplete in the introduction, especially about the research of differentially expressed genes (DEGs) involvement in periodontitis by computational tools or bioinformatic analysis. I suggest the authors to discuss the recent updates in the related field. Different feature weight calculation methods within a single algorithm result in divergent rankings of DEGs. I think we need consistency and comparable results.

Reply: Thank you for your suggestion. I quite agree with you. In fact, we used Degree, EPC, MNC and MCC algorithms to score DEGS, and finally screened out candidate genes conforming to the above algorithms. This approach is to evaluate the role of DEGS in diseases more objectively. Just like this article (PMID: 36960398).

7. Materials and methods section is relatively simple because of no detail about the analysis. The analysis methods and statistical parameters must be clearly emphasized (i.e, reasons for selecting the algorithms used, threshold values used in statistical analysis, etc.).

Reply: Thank you for your suggestion. In the part of materials and methods, I have added the specific process and parameters of relevant algorithms in detail.

Reviewer #2: 1. What do “LPS” and “LAPTM5, RAC2, LYN”represent in the Abstract section? What are their full names? When first mentioned, the author should provide their full names to help readers better understand.

Reply: Thank you for your suggestion. I have provided their full names in the abstract.

2.The authors should add a flowchart in the manuscript to show the process very well.

Reply: Thank you for your suggestion. I have added a flowchart in the manuscript.

3.The description of the result is quite simple and too short, especially the descriptions of “Identification of Common DEGs” and “The analysis of immune infiltration”. The authors should add some necessary sentences to describe these results.

Reply: Thank you for your suggestion. I have provided as detailed a description of the relevant results as possible in the Result part.

4.The labeling of Figures in the paper is quite small. The contents within the figures are unclear. The author needs to carefully revise and modify the figure.

Reply: Thank you for your suggestion. I have enlarged the words on my graphics as much as possible according to your suggestion.

5.The discussion of the deficiencies in current research is quite poor. The authors should discuss it as the future direction. ODE-based theoretical modeling studies on gene/protein signaling networks have been equally important for the study of understanding regulatory mechanisms and finding potential therapeutic targets in diseases (PMID: 35958114, https://doi.org/10.1016/j.chaos.2023.114328, and https://doi.org/10.1103/PhysRevE.108.064412). Would it be possible to discuss and cite these studies in conjunction with the conclusions of this paper?

Reply: Thank you very much for your constructive comments. I have carefully studied the relevant literature research provided by you. And added the discussion as the future direction.

6.Besides, the advancement of interaction prediction research in various fields of computational biology would provide valuable insights into genetic markers and related diseases. Important computational models in these fields should be discussed and cited. Some recommended studies are helpful (PMIDs: 36584603, 35817399, 36305458).

Reply: Thank you very much for your constructive comments. I have carefully studied the relevant literature research provided by you, which will benefit me a lot in my future research methods and methods. And in my discussion part, I introduce the model of relevant literature for discussion.

7.The authors should carefully check and unify the information of references. Some references lack the information of page number, such as refs [7] and [30].

Reply: Thank you for your suggestion. I have checked and unified the information of references

Best wishes

Luorong

---

## [Decision Letter · Decision Letter 1]

21 Mar 2024

PONE-D-23-39620R1Identification the hub genes of periodontitis and diabetes by bioinformatics and experimentPLOS ONE

Dear Dr. luo,

Thank you for submitting your manuscript to PLOS ONE. After careful consideration, we feel that it has merit but does not fully meet PLOS ONE’s publication criteria as it currently stands. Therefore, we invite you to submit a revised version of the manuscript that addresses the points raised during the review process.

We look forward to receiving your revised manuscript.

Kind regards,

Qi Zhao

Academic Editor

PLOS ONE

Journal Requirements:

Reviewers' comments:

Reviewer's Responses to Questions

**Comments to the Author**

1. If the authors have adequately addressed your comments raised in a previous round of review and you feel that this manuscript is now acceptable for publication, you may indicate that here to bypass the “Comments to the Author” section, enter your conflict of interest statement in the “Confidential to Editor” section, and submit your "Accept" recommendation.

Reviewer #1: (No Response)

Reviewer #2: All comments have been addressed

2. Is the manuscript technically sound, and do the data support the conclusions?

Reviewer #1: Yes

Reviewer #2: Yes

3. Has the statistical analysis been performed appropriately and rigorously? 

Reviewer #1: Yes

Reviewer #2: Yes

4. Have the authors made all data underlying the findings in their manuscript fully available?

Reviewer #1: Yes

Reviewer #2: Yes

5. Is the manuscript presented in an intelligible fashion and written in standard English?

Reviewer #1: Yes

Reviewer #2: Yes

6. Review Comments to the Author

Reviewer #1: 1.The workflow in supplementary material is unspecific and less organized. It's more like a stack of terminologies than a higher-level summary of the existing data and methodologies, please improve it. And this figure should be moved to the main body of revised manuscript.

2. The authors have already addressed most of the issues raised by the reviewer. But there are still some grammatical errors in the article, and the expression of some of the content is not clear enough. The authors need to check the manuscript carefully and make corresponding revision.

3. The authors should revise carefully corresponding to my previous comment #4. All the related paper should be cited.

Reviewer #2: The authors have addressed all the issues raised by the reviewer and can be accepted for publication in Plos one.

7. PLOS authors have the option to publish the peer review history of their article (what does this mean? ). If published, this will include your full peer review and any attached files.

**Do you want your identity to be public for this peer review?** For information about this choice, including consent withdrawal, please see our Privacy Policy .

Reviewer #1: No

Reviewer #2: No

---

## [Author Response · Author response to Decision Letter 2]

22 Apr 2024

Dear Editor:

I have provided the mapping data for the cell experiment section as Supporting Information. The original data about the analysis part of bioinformatic analysis exists in the GEO database as shared data, linked as follows (https://www.ncbi.nlm.nih.gov/) and the accession numbers as follows (GSE16134; GSE7014; GSE10334; GSE54675;).

Best wishes

Luorong

---

## [Decision Letter · Decision Letter 2]

25 Nov 2024

PONE-D-23-39620R2Identification the hub genes of periodontitis and diabetes by bioinformatics and experimentPLOS ONE

Dear Dr. luo,

Thank you for submitting your manuscript to PLOS ONE. After careful consideration, we feel that it has merit but does not fully meet PLOS ONE’s publication criteria as it currently stands. Therefore, we invite you to submit a revised version of the manuscript that addresses the points raised during the review process.

We look forward to receiving your revised manuscript.

Kind regards,

Simin Li, Ph.D.

Academic Editor

PLOS ONE

**Additional Editor Comments:**

Dear Dr. Rong Luo,

Thank you for submitting your revised manuscript "Identification the hub genes of periodontitis and diabetes by bioinformatics and experiment" (PONE-D-23-39620R2) to PLOS ONE. We have now received comments from reviewers regarding your revised submission, and I am writing to inform you that your manuscript requires further major revisions before it can be considered for publication.

While we acknowledge the improvements made in this revision, there are still significant concerns that need to be addressed. We have received contrasting recommendations from our reviewers, with one suggesting rejection and another recommending major revision. After careful evaluation of both reviews, I believe that your manuscript shows promise but requires substantial improvements to meet PLOS ONE's publication criteria.

When preparing your next revision, please provide a comprehensive response to all reviewer comments, ensuring that any changes are clearly highlighted in the manuscript. Please ensure that your conclusions are fully supported by your data and that all statistical analyses are appropriate and clearly described.

Please note that your revised version will be sent back to the reviewers for further assessment of how well you have addressed their concerns. We believe that carefully addressing these issues will significantly strengthen your paper and increase its potential impact in the field.

Please submit your revised manuscript within 30 days. If you anticipate any delay, please let us know your expected resubmission date by replying to this email.

We look forward to receiving your revised manuscript.

Best regards,

Academic Editor

PLOS ONE

Reviewers' comments:

Reviewer's Responses to Questions

**Comments to the Author**

1. If the authors have adequately addressed your comments raised in a previous round of review and you feel that this manuscript is now acceptable for publication, you may indicate that here to bypass the “Comments to the Author” section, enter your conflict of interest statement in the “Confidential to Editor” section, and submit your "Accept" recommendation.

Reviewer #1: All comments have been addressed

Reviewer #3: All comments have been addressed

Reviewer #4: (No Response)

Reviewer #5: All comments have been addressed

2. Is the manuscript technically sound, and do the data support the conclusions?

Reviewer #1: Yes

Reviewer #3: No

Reviewer #4: No

Reviewer #5: Yes

3. Has the statistical analysis been performed appropriately and rigorously? 

Reviewer #1: Yes

Reviewer #3: Yes

Reviewer #4: Yes

Reviewer #5: Yes

4. Have the authors made all data underlying the findings in their manuscript fully available?

Reviewer #1: Yes

Reviewer #3: (No Response)

Reviewer #4: Yes

Reviewer #5: Yes

5. Is the manuscript presented in an intelligible fashion and written in standard English?

Reviewer #1: Yes

Reviewer #3: No

Reviewer #4: Yes

Reviewer #5: Yes

6. Review Comments to the Author

**Reviewer #1: ** The author has already addressed all of the issues. I think the manuscript can be published in PLOS ONE.

**Reviewer #3: ** Thank you to the authors for the considerable amount of work done in the study "Identification of the hub genes of periodontitis and diabetes by bioinformatics and experiment," as well as for addressing some of the reviewers' comments during the revision process. However, from my personal perspective, I find it difficult to agree with the authors' results and conclusions. The main reasons are as follows:

The authors used the dataset GSE7014, which includes 10 DM1 biopsies, 20 DM2 biopsies, and 6 normal individuals' biopsies of skeletal muscle. The GSE16134 database includes a total of 120 patients undergoing periodontal surgery, each contributing with a minimum of two interproximal gingival papillae. This results in two major flaws for the study.

Firstly, Type 1 diabetes and Type 2 diabetes are diseases with completely different genetic and pathological mechanisms, making it entirely unacceptable to combine them. Secondly, there is a significant difference in gene expression between the interproximal gingival papillae and skeletal muscle tissues in the two datasets. Therefore, I regret to say that I cannot agree with your analysis results and conclusions.

**Reviewer #4:**  The manuscript titled "Identification of the hub genes of periodontitis and diabetes by bioinformatics and experiment" addresses an interesting intersection between periodontitis and diabetes mellitus, suggesting that shared molecular mechanisms could be explored via bioinformatics to identify key genes (LAPTM5, RAC2, and LYN). While the authors have provided a detailed bioinformatic analysis and experimental validation, there are significant concerns regarding the novelty and depth of the study:

The idea of identifying shared pathways between periodontitis and diabetes using bioinformatics is not entirely novel. Similar studies exist that delve into shared molecular mechanisms and immune responses in various diseases, including periodontitis and diabetes. The manuscript does not adequately highlight how its findings significantly advance the existing body of knowledge beyond what has been previously published.

While the study follows a well-known bioinformatic pipeline (DEGs identification, functional analysis, and protein-protein interaction network), it lacks innovation in methodology. The application of widely-used datasets (e.g., GSE7014 and GSE16134) without integrating more recent or diverse datasets from single-cell RNA sequencing (scRNA-seq) or other omics approaches limits the potential impact of the findings.

Furthermore, the study's use of in vitro validation, while helpful, does not explore functional assays beyond qPCR, and the validation itself remains quite basic.

The study relies on standard bioinformatic tools, but statistical robustness should be further scrutinized. For example, the selection of hub genes might benefit from additional cross-validation or more advanced machine learning techniques, which would add weight to the findings.

The authors have not integrated the latest available data on single-cell RNA sequencing for periodontitis or diabetes. Recent studies have shown the value of scRNA-seq in understanding cellular heterogeneity in diseases. The omission of these newer datasets weakens the study's relevance in the current scientific landscape.

Given these limitations, I would recommend rejecting the paper at this stage. The manuscript could benefit from incorporating more novel datasets, improving statistical validation, and considering recent developments in the field such as scRNA-seq for a more comprehensive analysis. Additionally, a more thorough discussion of how the results compare to and advance the current understanding of periodontitis-diabetes interactions would be necessary for future revisions.

**Reviewer #5: ** The manuscript, "Identification of the hub genes of periodontitis and diabetes by bioinformatics and experiment," has undergone thorough revisions, and I appreciate the authors' efforts in addressing the reviewers' feedback meticulously. The modifications significantly enhance the clarity and rigor of the study, particularly with the added flowchart, detailed descriptions of the methods and results, and improved figure readability. The authors have also provided a well-structured discussion, including the proposed future directions and acknowledgment of research limitations, making the study more robust. Moreover, the inclusion of computational models and ODE-based theoretical modeling studies as references aligns well with the study's objectives and brings additional depth to the discussion.

7. PLOS authors have the option to publish the peer review history of their article (what does this mean? ). If published, this will include your full peer review and any attached files.

**Do you want your identity to be public for this peer review?** For information about this choice, including consent withdrawal, please see our Privacy Policy .

Reviewer #1: No

Reviewer #3: No

Reviewer #4: No

Reviewer #5: **Yes: ** Shixiong Wei

---

## [Author Response · Author response to Decision Letter 3]

10 Dec 2024

Reviewer #1: The author has already addressed all of the issues. I think the manuscript can be published in PLOS ONE.

Reply: Thank you for your recognition.

Reviewer #3: Thank you to the authors for the considerable amount of work done in the study "Identification of the hub genes of periodontitis and diabetes by bioinformatics and experiment," as well as for addressing some of the reviewers' comments during the revision process. However, from my personal perspective, I find it difficult to agree with the authors' results and conclusions. The main reasons are as follows:

The authors used the dataset GSE7014, which includes 10 DM1 biopsies, 20 DM2 biopsies, and 6 normal individuals' biopsies of skeletal muscle. The GSE16134 database includes a total of 120 patients undergoing periodontal surgery, each contributing with a minimum of two interproximal gingival papillae. This results in two major flaws for the study. Firstly, Type 1 diabetes and Type 2 diabetes are diseases with completely different genetic and pathological mechanisms, making it entirely unacceptable to combine them. Secondly, there is a significant difference in gene expression between the interproximal gingival papillae and skeletal muscle tissues in the two datasets. Therefore, I regret to say that I cannot agree with your analysis results and conclusions.

Reply: Thank you very much for your valuable advice on my research. I rearranged our entire research program with your advice. In view of the “Type 1 diabetes and Type 2 diabetes are diseases with completely different genetic and pathological mechanisms”, I agreed that it was wrong to mix the two together. Therefore, I focus on DM2 as the main analysis object and PD analysis. Furthermore, for the point of “there is a significant difference in gene expression between the interproximal gingival papillae and skeletal muscle tissues in the two datasets”, I re-select GSE6751 (PD) and GSE15932 (DM2). The samples for these two data sets are from Peripheral Blood Mononuclear Cells (PBMC) samples. Through the above adjustments, I found more convincing results and conclusions than before. Thank you very much for your suggestions, which have laid a good foundation for my future work.

Reviewer #4: The manuscript titled "Identification of the hub genes of periodontitis and diabetes by bioinformatics and experiment" addresses an interesting intersection between periodontitis and diabetes mellitus, suggesting that shared molecular mechanisms could be explored via bioinformatics to identify key genes (LAPTM5, RAC2, and LYN). While the authors have provided a detailed bioinformatic analysis and experimental validation, there are significant concerns regarding the novelty and depth of the study: The idea of identifying shared pathways between periodontitis and diabetes using bioinformatics is not entirely novel. Similar studies exist that delve into shared molecular mechanisms and immune responses in various diseases, including periodontitis and diabetes. The manuscript does not adequately highlight how its findings significantly advance the existing body of knowledge beyond what has been previously published. While the study follows a well-known bioinformatic pipeline (DEGs identification, functional analysis, and protein-protein interaction network), it lacks innovation in methodology. The application of widely-used datasets (e.g., GSE7014 and GSE16134) without integrating more recent or diverse datasets from single-cell RNA sequencing (scRNA-seq) or other omics approaches limits the potential impact of the findings. Furthermore, the study's use of in vitro validation, while helpful, does not explore functional assays beyond qPCR, and the validation itself remains quite basic. The study relies on standard bioinformatic tools, but statistical robustness should be further scrutinized. For example, the selection of hub genes might benefit from additional cross-validation or more advanced machine learning techniques, which would add weight to the findings.

The authors have not integrated the latest available data on single-cell RNA sequencing for periodontitis or diabetes. Recent studies have shown the value of scRNA-seq in understanding cellular heterogeneity in diseases. The omission of these newer datasets weakens the study's relevance in the current scientific landscape.

Given these limitations, I would recommend rejecting the paper at this stage. The manuscript could benefit from incorporating more novel datasets, improving statistical validation, and considering recent developments in the field such as scRNA-seq for a more comprehensive analysis. Additionally, a more thorough discussion of how the results compare to and advance the current understanding of periodontitis-diabetes interactions would be necessary for future revisions.

Reply: Thank you for your valuable comments on my research. I re-adjusted my whole research according to the reviewer's opinion. Through this adjustment, I found a potential biomarker for type 2 diabetes and periodontitis -S100A9A. This finding is the first in the current study to propose S100A9 as a biomarker for both. The correlation was verified by PCR and WB experiments. Although I know my research still has shortcomings, please give me a chance, so that I can continue to have confidence in the follow-up experiment. Thank you very much for your advice. I think it is a very important experience in my academic career.

Reviewer #5: The manuscript, "Identification of the hub genes of periodontitis and diabetes by bioinformatics and experiment," has undergone thorough revisions, and I appreciate the authors' efforts in addressing the reviewers' feedback meticulously. The modifications significantly enhance the clarity and rigor of the study, particularly with the added flowchart, detailed descriptions of the methods and results, and improved figure readability. The authors have also provided a well-structured discussion, including the proposed future directions and acknowledgment of research limitations, making the study more robust. Moreover, the inclusion of computational models and ODE-based theoretical modeling studies as references aligns well with the study's objectives and brings additional depth to the discussion.

Reply: Thank you for your recognition.

---

## [Decision Letter · Decision Letter 3]

13 Feb 2025

Identification the hub genes of periodontitis and diabetes by bioinformatics and experiment

PONE-D-23-39620R3

Dear Dr. Luo,

We’re pleased to inform you that your manuscript has been judged scientifically suitable for publication and will be formally accepted for publication once it meets all outstanding technical requirements.

Kind regards,

Xiaozhe Han, D.M.D., Ph.D.

Academic Editor

PLOS ONE

Additional Editor Comments (optional):

Thanks for all the efforts to improve the manuscript. It is better organized now and contains new information to be shared with relative scientific community.

Reviewers' comments:

Reviewer's Responses to Questions

**Comments to the Author**

1. If the authors have adequately addressed your comments raised in a previous round of review and you feel that this manuscript is now acceptable for publication, you may indicate that here to bypass the “Comments to the Author” section, enter your conflict of interest statement in the “Confidential to Editor” section, and submit your "Accept" recommendation.

Reviewer #1: All comments have been addressed

Reviewer #4: All comments have been addressed

2. Is the manuscript technically sound, and do the data support the conclusions?

Reviewer #1: Yes

Reviewer #4: Yes

3. Has the statistical analysis been performed appropriately and rigorously? 

Reviewer #1: Yes

Reviewer #4: Yes

4. Have the authors made all data underlying the findings in their manuscript fully available?

Reviewer #1: Yes

Reviewer #4: Yes

5. Is the manuscript presented in an intelligible fashion and written in standard English?

Reviewer #1: Yes

Reviewer #4: Yes

6. Review Comments to the Author

Reviewer #1: The authors have made necessary revisions and has greatly improved the manuscript. I have no further comments.

Reviewer #4: I reviewed the manuscript carefully. The authors performed the revision properly. I have no further comments.

7. PLOS authors have the option to publish the peer review history of their article (what does this mean? ). If published, this will include your full peer review and any attached files.

**Do you want your identity to be public for this peer review?** For information about this choice, including consent withdrawal, please see our Privacy Policy .

Reviewer #1: No

Reviewer #4: **Yes: ** Yun Hak Kim

---

## [Editor Report · Acceptance letter]

PONE-D-23-39620R3

PLOS ONE

Dear Dr. luo,

I'm pleased to inform you that your manuscript has been deemed suitable for publication in PLOS ONE. Congratulations! Your manuscript is now being handed over to our production team.

Kind regards,

on behalf of

Dr. Xiaozhe Han

Academic Editor

PLOS ONE